# Antibiofilm Efficacies of Flavonoid-Rich Sweet Orange Waste Extract against Dual-Species Biofilms

**DOI:** 10.3390/pathogens12050657

**Published:** 2023-04-28

**Authors:** Suvro Saha, Thuy Do, Joanne Maycock, Simon Wood, Christine Boesch

**Affiliations:** 1School of Food Science and Nutrition, Faculty of Environment, University of Leeds, Leeds LS2 9JT, UK; 2School of Dentistry, Division of Oral Biology, Faculty of Medicine & Health, University of Leeds, Leeds LS2 9LU, UK

**Keywords:** citrus waste, citrus flavonoids, oral biofilm, antimicrobial, anticariogenic, caries

## Abstract

The current study evaluated the antibacterial properties of industrial sweet orange waste extracts (ISOWEs), which are a rich source of flavonoids. The ISOWEs exhibited antibacterial activity towards the dental cariogenic pathogens *Streptococcus mutans* and *Lactobacillus casei* with 13.0 ± 2.0 and 20.0 ± 2.0 mg/mL for MIC (minimum inhibitory concentration) and 37.7 ± 1.5 and 43.3 ± 2.1 mg/mL for MBC (minimum bactericidal concentration), respectively. When evaluated in a 7-day dual-species oral biofilm model, ISOWEs dose-dependently reduced the viable bacteria count, and demonstrated strong synergistic effects when combined with the anti-septic chlorhexidine (at 0.1 and 0.2%). Similarly, confocal microscopy confirmed the anti-cariogenic properties of ISOWEs, alone and in combination with chlorhexidine. The citrus flavonoids contributed differently to these effects, with the flavones (nobiletin, tangeretin and sinensetin) demonstrating significantly lower MICs and MBCs compared to the flavanones hesperidin and narirutin. In conclusion, our study demonstrated the potential of citrus waste as a currently underutilised source of flavonoids for antimicrobial applications, such as in dental health.

## 1. Introduction

Citrus fruit is globally in high demand compared to other fruit crops, with more than 143 million tons produced worldwide in 2019 [1,2]. Over half of the global citrus production are oranges, with sweet oranges (*Citrus sinensis*) being the most widely cultivated, followed by mandarins (*Citrus reticulata*), lemons (*Citrus limon*), limes (several species), and grapefruit (*Citrus paradisi*) [1]. Most of the sweet orange harvest is industrially processed, i.e., for juice production, resulting in the generation of huge amounts of solid waste that comprise up to 50% of the original processed whole fruit mass [3,4]. The juicing waste consists of the peel fractions (flavedo and albedo), the pulp (juice sac residue), rag (membranes and cores), and seeds [4]. The flavedo fraction contains the oil glands and accumulated carotenoids, whereas the albedo fraction is rich in secondary metabolites which include flavonoids, glucosides, pectin, and pectic enzymes [5]. In fact, the citrus peel waste contains higher amounts of the total phytochemical compounds compared to the edible portions; and these accumulated polyphenols in the outer layer of the peel protect the fruit from plant pathogens, insect attacks, and UV damage [6].

Although citrus waste may be rich in bioactive compounds, it is widely used as animal feed or sent for disposal in landfills [4]. There are constraints for citrus waste management due to both environmental and economic factors [7]. According to European regulations, citrus waste should not be used in landfills due to its low pH (3–4), high water content (approximately 80–90%), and high organic matter (approximately 95% of total solids) [8]. These organic compounds are associated with antimicrobial properties and can, therefore, damage the soil and water microflora when large amounts of untreated waste are released [9]. In addition, the high water and high soluble sugar content make the citrus waste easily perishable and fermentable, which contributes to environmental pollution [10].

Dental biofilm is composed of a diverse group of microorganisms that form on the dental surface. In a healthy oral environment, homeostasis of the dental biofilm is maintained among the polymicrobial communities, in contrast to oral dysbiosis, which can lead to the formation of a cariogenic biofilm [11]. One of the aetiologies for shifting healthy plaque to a caries-causing state is a sugar-rich diet [12]. *Streptococcus mutans*, one of the initial biofilm colonisers, has been identified as one of the primary cariogenic bacterial species due to its ability to metabolise a wide variety of sugars and synthesise extracellular polymers (EPS) with properties such as adhesion, acidogenicity, and aciduricity [13]. Apart from *S. mutans*, a number of *Lactobacillus* species have also been identified as caries-associated colonisers [11,13].

To date, chlorhexidine (CHX) has been considered the gold standard for the management of dental caries and gingival inflammation, and has had a significant clinical impact [14]. The long-term use of CHX, however, has drawbacks as it has caused denatured parotid enlargement, desquamation of the oral mucosa [15], and yellowish/brownish staining on the teeth and the dorsum of the tongue [16]. Given the CHX side effects as well as the rising incidence of antimicrobial resistance, there is a growing interest in the development of alternative antimicrobial strategies, e.g., involving natural bioactive compounds [17]. Polyphenols are a large group of plant secondary metabolites that demonstrate a range of bioactive properties [17,18]. Several citrus polyphenols have shown antioxidant, anti-diabetic, anti-inflammatory, antimicrobial, and anti-carcinogenic activities [18]. The antibacterial properties of sweet orange peel have been demonstrated in previous studies [19,20,21,22,23]. However, most of the research has been conducted on microorganisms in a planktonic state rather than a more realistic biofilm environment. This study aims to explore industrial sweet orange waste extracts (ISOWEs) as potential antibiofilm agents by testing them in a dual-biofilm model developed using *Streptococcus mutans* and *Lactobacillus casei* as the cariogenic pathogens.

## 2. Materials and Methods

### 2.1. Sample Preparation and Extraction

The sweet orange juicing waste was kindly provided by the company The Juice Executive (Kent, UK). After seed removal, the samples were cut into small pieces, frozen at −20 °C, and subsequently freeze-dried (Alpha 1-4 LD plus, Christ, Osterode, Germany) for 72 h at −10 °C. The freeze-dried samples were ground using an electric grinder (Nutribullet 600 series (NutriBulet, Los Angeles, CA, USA) followed by ball milling and sieving to achieve a final particle size of 90 µm and were stored at −40 °C in air-sealed bags. The extraction was carried out using the CEM Mars 6 closed microwave model (CEM Microwave Technology Ltd., Buckingham, UK) with a magnetic stirrer for adequate mixing during the extraction. In each extraction vessel, a 2 g sample was suspended in 20 mL of 70% aqueous ethanol and incubated for 5 min to absorb the solvent into the sample, followed by microwave-assisted extraction (MAE). MAE was carried out at two temperatures (70 and 90 °C) with maximum power of 300 W. The ramp time for both temperatures was 3 min with a temperature hold for 5 (A), 10 (B), and 15 (C) minutes. The extracts were labelled based on those conditions, as follows: 70 °C_A, 70 °C_B, 70 °C_C, 90 °C_A, 90 °C_B, and 90 °C_C.

Following the extraction, the vessels were immediately cooled on ice, then centrifuged for 10 min at 10,000× *g*, and the supernatants were filtered using Whatman No. 1 filter paper. The solvent was evaporated using a vacuum evaporator (Genevac EZ-2 Series, Genevac Ltd., Ipswich, UK) at 30 °C over 24 h, followed by freeze-drying for 24 h at −50 °C. The remaining ISOWEs were stored at −40 °C until use.

### 2.2. Characterization of the Extracts

#### 2.2.1. Determination of the Total Polyphenols and Total Flavonoids

The total polyphenol content (TPC) was measured using the Folin–Ciocalteu method in 96-well microtiter plates, according to Fernando et al. [24], with some modifications. Briefly, the industrial sweet orange waste extracts (ISOWEs) were prepared as 2 mg/mL stock solutions in ethanol with 0.1% DMSO and were further diluted with ethanol to fall within the gallic acid standard curve (0, 50–600 µg/mL). Absorbance measurements were performed at 765 nm using a Spark™ 10M multimode microplate reader (TECAN, Reading, UK). The standards and samples were assayed in triplicate and the results were expressed as g gallic acid equivalent (GAE)/100 g of dry weight (DW). The total flavonoid content (TFC) was determined using the aluminium chloride assay based on Diab et al. [6]. Catechin was used as a standard covering the range of 25–800 µg/mL. Briefly, 25 µL of the blank, standard, or sample, 10 µL of 5% NaNO_2_, and 100 µL of water were added to each well. After 5 min, 15 µL of 15% AlCl_3_ and 50 µL of 1 M NaOH were added, followed by the addition of 50 µL of water. The plate was incubated for 1 h and then read at 510 nm using the Spark™ 10M reader (TECAN). The results of the triplicate measurements were expressed as mean ± standard deviation in g catechin equivalent (CE)/100 g DW.

#### 2.2.2. Identification and Quantification of Flavonoids

An HPLC (LC-2010 HT) coupled with a 2020 quadrupole mass spectrophotometer (Shimadzu, Milton Keynes, UK) fitted with electrospray ionization (ESI-MS) was used to identify and quantify the flavonoid compounds in the ISOWE. This method was a modified version from Molina-Calle et al. [25]. A Phenomenex Germini C_18_ column (Macclesfield, UK) (4.6 × 250 mm, 5 µm) was used with the following mobile phases: a 0.5% aqueous formic acid (A), a 0.5% formic acetonitrile (B), and a flow rate of 0.2 mL/min. The solvent gradient was as follows: 0–10 min, 5% B; 10–20 min, 5–20% B; 20–30 min, 20–30% B; 30–40 min, 30–50% B; 40–50 min, 50–60% B; 50–60 min, 60–70% B; 60–80 min, 80–90% B; 80 min 20% B; 80–100 min, 5% B. The temperature of the column oven was set to 35 °C and the injection volume was 10 µL. Both single ion monitoring (SIM) and scans were used in negative mode. The flavonoids were monitored at 285 nm. The stock solutions of the ISOWE were initially prepared at 800 mg/mL in ethanol (3% DMSO) and passed through a sterile 0.45 µm membrane filter. The flavones were detected at 400 µg/mL, and the concentrations of 40 and 2 µg/mL were used for the quantification of narirutin, quercetin, and hesperidin, respectively. The external standards (hesperidin, narirutin, quercetin, sinensetin, nobiletin, tangeretin; all obtained from Extrasynthesis (Genay, France)), were used for the comparison in a range of 12.5–400 µg/mL.

#### 2.2.3. Determination of the Antioxidant Capacity

Antioxidant capacity (AOC) was measured using Trolox equivalent antioxidant capacity (TEAC) and oxygen radical absorbance capacity (ORAC) assays. Trolox was used as standard for both assays, which were performed in 96-well microtiter plates in triplicate. The TEAC assay was adapted from Han et al. [26]. Briefly, 10 µL of the standard or sample was mixed with 300 µL of the ABTS radical working solution, followed by incubation for 6 min in the dark at room temperature. The absorbance was read at 734 nm using the Spark™ 10M plate reader (TECAN). The extracts were tested at 600 µg/mL and the results were expressed as mean ± standard deviation in mM Trolox equivalent (TE)/g DW. The ORAC assay was adapted from Huang et al. [27] with modifications. Briefly, 25 µL of blank, standard, or sample and 150 µL of the fluorescein working solution were added into the wells of a 96-well plate followed by incubation at 37 °C for 30 min. Then, 25 µL of the AAPH solution (153 mM) was rapidly added to each well and immediately placed in the plate reader. The fluorescence intensity was recorded every min for 150 min at 37 °C with excitation at 485 nm and emission at 520 nm. The net area under the curve (AUC) was calculated for the blank, Trolox standards and extracts by subtracting the AUC of standards/extracts from the AUC of the blank.

### 2.3. Antibacterial Experiments

*Streptococcus mutans* NCTC 10449 and *Lactobacillus casei* 2104A were cultured aerobically using brain heart infusion (BHI) at 37 °C. Both microorganisms were streaked onto BHI agar plates every two days for the entire experimental period. To prepare the bacterial inoculum, overnight cultures in a BHI broth were grown to a ~10^8^ CFU and diluted to 1% solutions for the MIC/MBC experiments. Sucrose (2%) was added to the BHI broth (sucrose fortified the BHI broth) to imitate a diet rich in fermentable carbohydrates [12].

#### 2.3.1. Antibacterial Properties of the Extracts and Test Compounds

In order to determine the minimum inhibitory concentration (MIC), the test compounds were serially diluted in a sucrose-fortified BHI medium and incubated for 24 h in the presence of both bacteria (1%) in flat-bottomed 96-well plates in a final volume of 200 µL. The stock solutions of the ISOWE (250 mg/mL) and CHX (2%) were initially prepared in BHI broth, containing 4% and 2% of DMSO, respectively. The individual flavonoids (hesperidin, narirutin, quercetin, sinensetin, nobiletin, and tangeretin) were prepared as 2 mg/mL stock solutions using 2% DMSO. The respective DMSO concentrations were required to dissolve the test compounds. In addition, both positive and negative controls were prepared with the highest concentration of DMSO to confirm whether the antimicrobial effects were solely due to the test compounds. The lowest concentration of the test compound that inhibited the visible growth of the organism, assessed as medium turbidity, was considered to be the MIC. The minimum bactericidal concentration (MBC) is the lowest concentration of a compound that causes bacterial cell death, resulting in a lack of colony formation after plating onto any non-selective agar medium. The MBC was determined using the ‘spot method’ based on Suppi et al. [28] with modifications. With this method, 4 µL of a suspension was spotted (in triplicate from each well) onto BHI agar plates and incubated at 37 °C aerobically for 48 h. The MIC and MBC assays were performed in two steps, as outlined in Figure 1.

#### 2.3.2. Cariogenic Dual-Species Biofilm Model

The biofilm model was modified from Jeon et al. [29] and conducted under anaerobic conditions at 37 °C. Vertically placed sterile hydroxyapatite (HA) discs were incubated with sterilised saliva in an Eppendorf tube for 24 h. The saliva was donated by the experimenter and prepared according to Naginyte et al. [30] to precondition the HA discs. Subsequently, the saliva was removed and replaced by 200 µL biofilm growth medium (BGM), which was prepared by diluting artificial saliva (600 mL/L) with basal media (200 mL/L), following the addition of heat-inactivated foetal bovine serum (200 mL/L) to simulate the gingival environment. The artificial saliva consisted of the following (g/L): porcine gastric mucin (2.5), NaCl (0.381), KCl (1.114), KH_2_ PO_4_ (0.738), ascorbic acid (0.002) as well as urea (9 mM) and L-arginine (5 mM). The basal media consisted of (g/L) protease peptone (10.0), tryptose peptone (5.0), yeast extract (5.0), L-cysteine hydrochloride (0.5), haemin (0.0002), and menadione (0.00004). Sucrose (2%) was added to the BGM to prepare a sucrose-fortified BGM (SFBGM). A bacterial inoculum was added for both microorganisms as 2%. After 24 h, the HA discs were gently washed three times with PBS to remove the loose bacterial cells, exposed to sucrose for 20 min by introducing 200 µL of SEBGM into the tube, followed by washing the discs with PBS and incubation in the BGM. The sucrose exposure was performed three times a day in 6 h intervals for 7 days to form a cariogenic biofilm.

The cariogenic biofilm was treated with respective solutions for 4 days. Each day, the treatment was performed two times at an interval of 12 h with a treatment duration of 1 min. The discs were washed with PBS and incubated in the BGM after each treatment. On the day after the 4-day treatments, the bacterial count reduction was determined by the number of viable bacteria (CFU/mL) and the viable/dead bacterial cell stain visualised using confocal laser scanning microscopy (CLSM). The most potent ISOWE was selected for biofilm experiments based on their lowest MIC, and was used as 2×, 4×, and 6× MIC. CHX was used at 0.1% and 0.2% and in combination with ISOWE. The exposures with no sugar or sucrose were used as control conditions.

#### 2.3.3. Viable Bacterial Count

After cleaning with PBS, the HA discs were transferred into sterile Eppendorf tubes containing 1 mL of RTF and sterile glass beads and vortexed for 1 min to disperse the biofilm. Then, 100 µL of the biofilm suspension was transferred to a new tube containing 900 µL of RTF and further diluted in 10-fold serial dilutions (10^−1^ to 10^−10^). A volume of 100 µL of each dilution was spread onto MRS, Rogosa, and BHI agars to determine the viable counts of *S. mutans*, *L. casei*, both alone and in combination. The MSA (mitis-salivarius agar) was modified according to Liliana et al. [31] by adding sucrose (20%), potassium tellurite (1%), and bacitracin (0.3 U/mL) to suppress the growth of *L. casei*. The bacitracin concentration was determined in preliminary experiments (0.2–0.6 U/mL).

### 2.4. Bio-Imaging

#### 2.4.1. Confocal Laser Scanning Microscopy (CLSM)

CLSM was used to visually determine the bacterial cell viability using the Live/Dead BacLight bacterial viability kit. The HA discs were washed with PBS and a 1:1 mixture of SYTO 9 and propidium iodide stain (0.6 µL) was applied in 200 µL of sterile PBS and incubated for 20 min in the dark at room temperature, followed by gentle washing with PBS before being observed using confocal microscopy. The excitation/emission wavelengths were 480/500 and 490/635 nm for SYTO 9 and propidium iodide, respectively.

#### 2.4.2. Scanning Electron Microscopy (SEM)

For SEM, the discs were dipped into PBS to wash the loosely adhered bacterial cells and then dipped in 2.5% glutaraldehyde for 3 h at room temperature to fix the sample. The discs were washed three times with PBS and subsequently dried by sequential dipping in 10, 30, 50, 70, 90, and 100% ethanol for 15 min each. The 100% ethanol step was repeated three times followed by air drying. Then, the discs were sputter-coated with gold for 80 s and examined under high vacuum at 10.00 kV with a working distance of 9.00 to 10.00 mm.

### 2.5. Statistical Analysis

The data were reported as mean ± standard deviation (*n* = 3) and the graphs were plotted using GraphPad Prism version 9.3.1 (Boston, MA, USA). All the data were analysed using two-way ANOVA with a subsequent Tukey’s multiple comparisons test to detect significant differences between the experimental groups. The threshold for significance was set at *p* < 0.05. The correlation among the TPC, TFC, individual flavonoids, AOA, and MIC and MBC assays was expressed as Pearson’s correlation coefficients using the same software.

## 3. Results and Discussion

### 3.1. Total Polyphenol, Flavonoid Content, and Antioxidant Activities of Sweet Orange Extracts

There is increasing interest in the utilization of agriculture waste streams towards the recovery of bioactive fractions/high value compounds and the development of downstream applications. In the current project, juicing waste from the orange processing industry was used as the starting material with MAE being applied for the extraction of polyphenols. Microwave-assisted extraction is considered a green extraction method and has been reported to improve the extraction in terms of yield and uniformity. It is considered more economic due to its reduced requirements for solvent volume, time, and energy, compared to other extraction processes [32]. In general, a range of factors can influence extraction outcome, such as extraction solvent, sample/solvent ratio, particle size, temperature, and duration [6,33,34]. Hence, a wide range of results regarding TPC, TFC, individual flavonoids, and AOA of sweet orange peels have been reported in previous studies [32,34,35,36].

For the present study, conducted at different temperatures (70, 90 °C) and durations (5, 10, 15 min), overall similar TPC values for all extractions were achieved, ranging between 4.48 ± 0.57 and 5.59 ± 1.01 g GAE/100 g DW, except at 90 °C with extraction durations of 10 and 15 min, which resulted in significantly lower TPCs with 3.38 ± 0.17 and 2.46 ± 0.29 g GAE/100 g DW, respectively (Figure 2a). The highest flavonoid contents were achieved at 90 °C with a 5 min extraction, yielding 2.64 ± 0.20 g CE/100 g DW (Figure 2b). Overall, all the extractions at 70 °C did not differ from each other for both the TPC and TFC, but a prolonged extraction duration (10 and 15 min) at a higher temperature (90 °C) decreased the extraction yield.

Various studies on the ISOWE, including Petrotos et al. [35], reported ranges of the TPC and TFC between 0.004–4.92 g GAE/100 g DW and 0.2–3 g QE/100 g DW, respectively. Petrotos et al. [35] described optimised parameters for MAE of solid waste from the orange juice producing industry using a lab-scale microwave extractor in a water-based extraction approach. The maximum TPC and TFC were 1.86 g GAE/100 g DW and 0.18 g QE/100 g DW, respectively, with the corresponding optimal extraction parameters temperature and time for TPC (80 °C and 30.5 min) and TFC (79.6 °C and 89.3 min). More recently, this group [34] optimised the ISOWE extraction method using an upgraded industrial-scale microwave extractor, which increased TPC and TFC to 3.77 g GAE and 0.53 QE per 100 g DW, respectively. The increased MAE power and extraction time had a positive impact on the TPC and TFC, but after reaching a maximum, it decreased due to the sensitivity of the phytochemical compounds. In comparison, the higher TPC and TFC contents that were found in the current study were most likely due to the use of ethanol for the extraction. Ethanol is considered one of the best extraction solvents as it is a good microwave absorber and has hydrophilic (OH group) and hydrophobic (hydrocarbon) ends [6]. The presence of these moieties helps to extract both polar and non-polar compounds. Additionally, the addition of water to ethanol increases the extractability of polyphenols.

Figure 2c,d show the results of the antioxidant activities in the extracts that were determined using TEAC and ORAC assays. At 70 °C, AOA (TEAC) was the highest after 15 min (66.5 ± 4.2 mM TE/g), whereas at 90 °C, the highest values were achieved after a 5 min extraction (77.8 ± 2.2 mM TE/g), with the latter being aligned with the TPC and total flavonoid content. A prolonged extraction time of 15 min at 90 °C resulted in the lowest values, which was even more evident in the ORAC assay results, where the lower AOA was evident already at a 10 min extraction time. In addition, there was no difference in the ORAC values for all the extracts obtained at 70 °C, which were not dissimilar to the ORAC results after a 5 min extraction at 90 °C (208.8 ± 18.6 mM TE/g). Although the TPC and TFC reported by Liew et al. [37] were lower for similar samples, their ORAC value (680 mM TE/g) was much higher compared to our study.

A reason for the differing outcomes compared to the literature may also relate to the sample processing. In the present study, freeze-drying was applied to remove the remaining moisture from the material which was in contrast to others who used heat drying that can cause an oxidative degradation of the polyphenols and may be an explanation for the deviating extraction results [34,35].

Many studies [38,39,40] reported a strong correlation between the total polyphenol, flavonoid compounds, and antioxidant capacity, which was not observed in the present study. Both the TPC and TFC of the different extracts correlated differently with the AOA (ORAC and TEAC). Strong positive correlations were found between TPC and ORAC of at 70 °C, 10 min (r = 0.9804; *p* < 0.05) and the TPC and TEAC at 90 °C, 15 min (r = 0.9667; *p* < 0.05). In case of TFC, a strong correlation was observed between the TEAC at 70 °C (10 min) (r = 0.9538; *p* < 0.05) and 90 °C (15 min) (r = 0.9094; *p* < 0.05).

### 3.2. Identification and Quantification of Flavonoid Compounds in Sweet Orange Peel Extracts

The flavonoids that were identified in the ISOWE were the flavanones narirutin, hesperidin, the flavonol quercetin, and the flavones sinensetin, nobiletin, and tangeretin, as indicated in a representative chromatogram (Figure 3). Hesperidin was found to be dominant in all the extract samples in the concentrations ranging from 1.42–2.76 g/100 g, with the highest concentrations of 2.69 and 2.76 g/100 g achieved after a 5 min and 10 min extraction at 70 °C, respectively (Table 1). All the other extracts demonstrated significantly lower hesperidin values, emphasizing that higher temperatures in combination with a prolonged incubation time was not beneficial towards the extractability of hesperidin. In contrast, the concentrations of the other flavonoids were much lower in all the extracts with values below 0.5 g/100 g. While there was no difference for quercetin in all the extract conditions, the narirutin and sinensetin extraction was higher after 5 and 10 min at 70 °C as well as after 5 min at 90 °C. The findings at 90 °C were particularly in line with the total polyphenol and flavonoid content (Figure 2). The lowest concentrations were determined in case of nobiletin and tangeretin, which showed values around 0.1 g/100 g in all the samples except those that were extracted for 10 min or longer at 90 °C.

Overall, our results, including the predominance of hesperidin, were in line with other studies on sweet orange peel that reported similar flavonoid patterns [34,35,41]. An exception might be the study of Inoue et al. [36], which used a response surface methodology-based approach for optimisation of extraction for hesperidin, narirutin, and nobiletin. The authors were able to achieve values of 5.86, 1.31, and 0.02 g/100 g DW, respectively, using a closed MAE system with 70% aqueous ethanol as the solvent at 140 °C and a 7 min extraction time. In the current work, it was not possible to apply temperatures beyond 100 °C due to the safety restrictions given by the equipment manufacturer. While the hesperidin and narirutin quantities were significantly higher compared to the present study, the extracted amount of nobiletin was much lower.

Apart from the different extraction conditions, findings of differing amounts of citrus flavonoids and their proportions may have been an expression of other factors, such as variation in the starting material, e.g., varietal differences, the type of sample, and the pressing method, leading to different proportions of juice and peel in the starting material, which reflects the polyphenol content, composition, and AOA properties. Apart from the polyphenols, other components such as ascorbic acid may have been present in the samples, i.e., the juice fraction, and were reported to rescue sulphur radicals and quench reactive oxygen species (ROS) [37]. In any case, citrus waste is considered an important flavone source compared to other plants, albeit in low concentrations [42,43].

### 3.3. Antimicrobial Effects of ISOWE and CHX

The flavonoid-rich ISOWE, as well as the individual flavonoids present in the extracted samples, were investigated for their potential to inhibit the growth and viability of the cariogenic bacterial species *S. mutans* and *L. casei* (Figure 4). The MIC and MBC of 70 °C_B against *S. mutans* were 13.00 ± 2.00 and 37.67 ± 1.53 mg/mL, respectively, and this extract did not show any significant difference compared to 70 °C_C and 90 °C_A (Figure 4a). The statistical significance of the MIC and MBC of the ISOWE against *L. casei* (Figure 4b) were similar to that of *S. mutans*. However, *S. mutans* showed more susceptibility towards the ISOWE compared to *L. casei*. In contrast, the individual flavonoids greatly varied in the range of their MICs and MBCs (Figure 4c,d), with tangeretin and sinensetin being the most potent, followed by nobiletin and quercetin. The least potent were hesperidin and narirutin, for which only the MIC could be determined in this present study. Similar to the ISOWE, *S. mutans* showed a higher sensitivity to all detected flavonoids compared to *L. casei*.

Chlorhexidine (CHX), a well-known and widely used antibacterial agent in dental care settings, showed superior efficacy in reducing growth and viability of both bacterial strains, with MIC values below 10 µg/mL and an MBC around 20 µg/mL. Although the MIC for CHX against *S. mutans* and *L. casei* was not significantly different, the MBC for *S. mutans* was higher than for *L. casei*.

The bactericidal kinetics experiment was carried out using the pre-determined MBC concentrations for 70 °C_B and 90 °C_A against *S. mutans* and *L. casei*. The 90 °C_A extract had the lowest MIC value among the respective temperature group.

The bactericidal kinetics for the selected extracts were the same for both organisms, as well as ISOWE and pure flavonoids, which showed no difference in their growth inhibitory properties. Although CHX was more effective than ISOWE when used separately, a synergistic effect was observed when CHX was combined with the selected extracts. The experiments were carried out using different doses of both CHX and the extracts, and no difference was observed in the growth inhibition for either bacteria.

CHX acts as both a bacteriostatic and bactericidal agent depending on low or high concentrations, respectively [44]. According to Al-khalifa et al. [45], the MIC and MBC of CHX against *S. mutans* were 0.02 and 0.9 µg/mL, respectively. Compared to the present study, the required CHX concentrations for the MIC and MBC (9 and 26 µg/mL, respectively) against *S. mutans* were higher. However, Al-khalifa et al. [45] reported a bactericidal time (<3 h) that was less than the current study (14.43 ± 0.58 h). According to Niu et al. [46], a 0.12% CHX solution could kill all the *S. mutans* cells within 5 min, and according to Wang et al. [47], the planktonic state of *S. mutans* (1 and 4 µg/mL for the MIC and MBC, respectively) was more susceptible towards CHX compared to *L. casei* (2 and 8 µg/mL for the MIC and MBC, respectively). Chung et al. [48] also reported a similar susceptibility, but the required CHX concentration for the MIC (31.3 µg/mL) and MBC (62.5 µg/mL) for *L. casei* were higher compared to the current study. On the contrary, we found *L. casei* to be more sensitive in respect to the MBC (16 µg/mL) for the CHX treatment, but no significant difference for the MIC of CHX compared to *S. mutans*. A higher sensitivity to CHX was observed for *Lactobacillus* sp. compared to *S. mutans*, as described by Gondim et al. [49]. The same MIC (70 µg/mL) and MBC (150 µg/mL) concentrations for CHX were reported against *S. mutans* and *L. acidophilus* [50]. In a comparison between *L. casei* and *L. acidophilus* for CHX sensitivity, the MBC for *L. casei* required almost double the concentration of CHX but both MICs were the same [47,48]. According to the study conducted by Akca et al. [51], *S. mutans* showed more resistance (both an MIC and MBC of 16 µg/mL) compared to *L. acidophilus* (4 and 8 µg/mL).

The antimicrobial efficacies of ISOWE could be attributed to its flavonoid content as the pure flavonoids showed no significant differences in the inhibition of pathogens compared to ISOWE. Unlike CHX, the ISOWE and pure flavonoids showed an increased effectiveness against *S. mutans* compared to *L. casei*. The water content of the orange peel extract (from fresh fruit) exhibited no antimicrobial activity, but the ethanol extract inhibited (MIC with 2.50 mg/mL) *S. aureus* [52]. Musa et al. [53] also reported antimicrobial activity of orange peel extract against *S. aureus*. The MICs of ISOWE against *S. mutans* and *L. casei* reported in the present study were higher compared to either of these previous studies [52,53]. The MICs and MBCs of 70 °C_B, 70 °C_C, and 90 °C_A did not differ significantly, however, 70 °C_B was selected in the antibacterial experiments since its lower temperature and extraction time consists a more sustainable option. The mechanism for the ISOWE’s biofilm growth reduction might have involved quorum sensing. Truchado et al. [54] reported that the aqueous extract of fresh orange peel inhibited the formation of *Yersinia enterocolitica* biofilm by modulating quorum sensing without affecting the bacterial growth. The mechanisms for the antibiofilm activity of the ISOWE is still unclear and further studies are needed.

Although flavanones (hesperidin and narirutin) were the predominant compounds in ISOWE, both *S. mutans* and *L. casei* showed the least susceptibility compared to all detected flavonoid compounds. Within the given maximum range (2 mg/mL) of the hesperidin and narirutin concentrations, ISOWE did not show bactericidal properties. Nomura et al. [55] also reported the inactivity of hesperidin (10 mg/mL) against *S. mutans*. Karuppiah et al. [56] established the antibiofilm potential of hesperidin (100 µg/mL) against methicillin-resistant *S. aureus*. It was capable of inhibiting the autoaggregation, synthesis of lipase, hemolysin, and staphyloxanthin, and downregulating the genes that help with adhesion without affecting the bacterial growth [56]. Hesperidin was also reported to enhance the remineralisation process and, thereby, reduce the susceptibility of dentin demineralisation [57,58]. A previous study [59] reported that the MIC of quercetin against *S. mutans* was 1.5 mg/mL, which was higher compared to the present study. This concentration of quercetin (1.5 mg/mL) was able to reduce 75% of the biofilm and 45% of the reducing sugar concentration in a 36-h matured *S. mutans* biofilm [59]. Smullen et al. [60] reported that quercetin (1.5–50 µg/mL) significantly inhibited the glucan formation of *S. mutans* in a dose-dependent manner. According to Jaisinghani et al. [61], quercetin did not show any antimicrobial activity against *L.* casei but it was effective against *S. aureus* (MIC and MBC with 20 and 50 µg/mL, respectively) and *E. coli* (MIC and MBC with 400 and >500 µg/mL, respectively). In the present study, the detected flavones (tangeretin, nobiletin, and sinensetin) exhibited a stronger antimicrobial activity against both pathogens compared to the other detected flavonoids. Among the three detected flavones, tangeretin and sinensetin were more susceptible compared to nobiletin. Yi et al. [62] reported that the antimicrobial effectiveness of hesperidin was higher compared to nobiletin and tangeretin, and there was no significant difference in the MIC between these two flavones. Yao et al. [63] evaluated the antimicrobial effects of nobiletin and tangeretin against *P. fluorescens* and *P. aeruginosa*. According to this study, tangeretin was more effective against both microorganisms compared to nobiletin [63]. Sinensetin (100 µg/mL) was reported to inhibit the biofilm of *E. coli* and *Vibro harveyi* without inhibiting the growth of both bacteria [64]. In the same study, the effect of sinensetin and nobiletin on the mobility of *E. coli* was demonstrated and nobiletin was more effective compared to sinensetin [64]. The antibiofilm capacity of sinensetin was stronger than hesperidin, quercetin, naringin, and naringenin against the *E. coli* biofilm [65]. The antimicrobial mode of action of nobiletin and tangeretin was the disruption of the bacterial cell permeability and the hinderance of the protein synthesis, leading to cell pyknosis and death [63,66]. The antibiofilm mode of action for flavonoids involves damaging the cytoplasmic membrane, suppressing nucleic acid synthesis, downregulating energy metabolism, inhibiting quorum sensing, and blocking the active site for bacterial attachment to the tooth surface [65,67]. The inconsistency in the antimicrobial activity observed in this study was likely due to the different methods of the assay, inoculum concentration, and polyphenol sources. Carmona and Pereira [68] and Cravotto et al. [69] reported that the crude extract had more potential as an antimicrobial agent compared to single pure bioactive compounds due to multiple antimicrobial mechanisms of the bioactive compounds present in a crude extract. In addition, extracting the pure compounds from ISOWE may be costly, which will ultimately influence the price of the end product(s).

### 3.4. Effect of the ISOWE on a 7-Day Dual-Species Cariogenic Biofilm

A human lifestyle that includes a sugar-rich diet, smoking, alcohol consumption, antibiotic misuse, and poor oral hygiene can influence the stability of the oral biofilm [70]. Among those factors, the frequent consumption of a sugar-rich (mostly sucrose) diet is considered to be the primary factor for shifting the normally symbiotic microbial population towards that of a caries-associated population [70]. Salli and Ouwehand [71] reported that the majority of biofilm models involve constant sucrose exposure but this is rarely common in the oral environment. Therefore, in this study, a cariogenic model was developed by exposing a 2% sucrose enriched medium for 20 min three times a day at 6 h intervals between the exposures. The procedure of these sucrose exposure was continued for 7 days. The HA discs were used as the substrata for the biofilm development and were positioned vertically in the Eppendorf tubes to avoid sedimentation of the microorganisms to the disc and to mimic the natural oral biofilm development. The biofilms that developed in the presence of sucrose enriched media were considered to be the cariogenic biofilm in this study. The metabolism of dietary sucrose by the oral microorganisms caused a decrease in the biofilm pH and promoted a thicker architecture of the biofilm, hindering the salivary buffering which led to the accumulation of lactic acid on the tooth surfaces and increased enamel demineralisation. The behaviour pattern of any microorganism in the planktonic state was not comparable to that of a biofilm structure. The clinical significance associated with biofilms is the dramatic increase in the virulence properties of the microorganisms and the increased antibiotic resistance. Biologically, these cariogenic biofilms drastically alter their physicochemical characteristics, strengthening their adhesion properties and making them resilient to salivary buffering and antimicrobials [72].

Among all the ISOWEs, 70 °C_B was selected to explore its antibiofilm activity since the MIC data and the kinetic study showed no significant difference between 70 °C_B and 90 °C_A. The 2×, 4×, and 6× MIC of the 70 °C_B concentrations were chosen based on the dose required for *L. casei,* which was higher compared to *S. mutans*. The CHX concentrations in many commercial mouthwash products range between 0.1 to 0.2 [73], hence these two CHX concentrations were used in this study to compare the efficacy of this extract. Figure 5 indicates the antibiofilm efficacy of the ISOWEs (40, 80, 120 mg/mL), CHX, and a combination of the two. The MSA and RA media selectively supported the growth of *S. mutans* and *L. casei*, respectively. Both microorganisms grew on the BHA media. Therefore, the CFU/mL from the BHI media represented the total number of bacteria in the sample.

In the presence of sucrose, the bacterial viable count (Figure 5c) increased, and the architecture of the biofilm was very thick and condensed (Figure 5b) compared to the biofilm developed in the absence of sucrose (Figure 5a). The viability count (Figure 5c) and confocal images (Figure 5d–k) showed that the antibacterial activity of both CHX and the ISOWE was proportional to the dose. The CLSM images for the non-treated cariogenic biofilm (Figure 5d) showed a higher proportion of green colour (viable bacterial cells) compared to the images for the different treatments. The CFU on the BHI with 0.2 and 0.1% CHX were 12–14 × 10^3^ and 25–28 × 10^3^, respectively. The CFU/mL on the BHI, MSA, and RA media for the ISOWE with a 120 mg/mL concentration were 12–22 × 10^8^, 12–71 × 10^7^, and 40–80 × 10^7^, respectively. The treatment with all three concentrations of the ISOWE against the cariogenic biofilms significantly reduced the bacterial viability from all the media compared to the control cariogenic biofilm. The CHX concentrations showed better anticariogenic properties compared to the ISOWE. The number of viable bacteria on the BHI media (65–83 × 10^2^) was markedly reduced after treating the dual-species cariogenic biofilm model with the combination of CHX (1%) and ISOWE (120 mg/mL). This combination showed better antibacterial activity compared to the treatment using 0.2% CHX alone. Similar trends were also observed with the other two media. The antimicrobial synergy through the combination of 0.1% CHX and the 120 mg/mL ISOWE is shown in Figure 5j where the biofilm was thinner with more dead bacterial cells compared to the biofilm treated using 0.2% CHX alone.

Both CHX and the ISOWE showed dose-dependent antibacterial effects. The dose-dependency of CHX was reported by Akca et al. [51], demonstrating MBC of 1024 µg/mL against a 7-day cultured *S. mutans* biofilm. In a 24-h matured *S. mutans* biofilm, CHX at concentrations of 1 × 10^3^ and 2 × 10^3^ inhibited approximately 50% and 90% of the biofilm formation, respectively [46]. The confocal images indicated that the proportion of dead cells (red stain) increased with higher CHX concentrations, and the opposite trend was noticed for the intensity of the green colour (live bacterial cells) [46]. Wu et al. [74] reported that 0.12% of the CHX concentration treatment for 1 min on 72-h matured biofilm of *S. mutans* did not show any significant difference in the viable bacterial count (CFU/mL) compared to the control group (CFU/mL with more than 10^7^). However, 0.12% CHX significantly reduced the viability of *L. acidophilus* in the 72-h matured biofilm [74].

Apart from the decreased effectiveness of CHX against matured plaque due to biofilm structure and associated properties, including increased antimicrobial resistance, the daily and/or longer period use of CHX causes adverse effects, including tooth and tongue staining, a burning sensation in the mouth, calculus formation, and ecological imbalance due to the reduction in the number of beneficial commensals [14]. To our knowledge, no other study has investigated the antimicrobial potential of sweet orange peel (both from industrial waste and fresh fruit) against a cariogenic biofilm. The flavonoids that were detected in this study demonstrated an antibiofilm capacity in other reports [55,56,59,60]. The antibacterial effectiveness of CHX was higher compared to the ISOWE. However, combining CHX and the ISOWE significantly shortened the necessary time for bactericidal kinetics and the bacterial viability from the biofilm compared to CHX and ISOWE by themselves. The combination of 0.1% CHX and the 120 mg/mL ISOWE was more effective as an antibacterial agent compared to 0.2% CHX alone. The synergism between a lower concentration of CHX and the ISOWE could help in lowering the side effects of CHX.

In this study, the antimicrobial activity of the ISOWE was observed against a biofilm model composed of two selected bacteria that were associated with caries. It would be important to explore the effects of ISOWE on a diverse and more complex group of oral microorganisms, including the pathogens and commensals associated with health. Moreover, the crude citrus waste extracts likely also contain other compounds, such as other polyphenols, essential oils, and pectins which were not the focus of this study.

## 4. Conclusions

In summary, microwave-assisted extraction of flavonoids from sweet orange peel waste resulted in extracts with distinct polyphenol profiles and antioxidant capacities that demonstrated great potential for affecting the growth and viability of oral bacterial strains associated with caries. These effects were strongest for the polymethoxylated flavonoids tangeretin, nobiletin, and sinensetin. Notably, the flavonoid-rich ISOWE counteracted the biofilm formation, and displayed synergistic effects in combination with chlorhexidine. Our results indicate the potential of the citrus waste bioactive compounds as antibacterial agents and highlight the need for further research in order to optimise the drug combination treatment and test the efficacy in more complex environments in vitro and in in vivo clinical settings.

## Figures and Tables

**Figure 1 pathogens-12-00657-f001:**
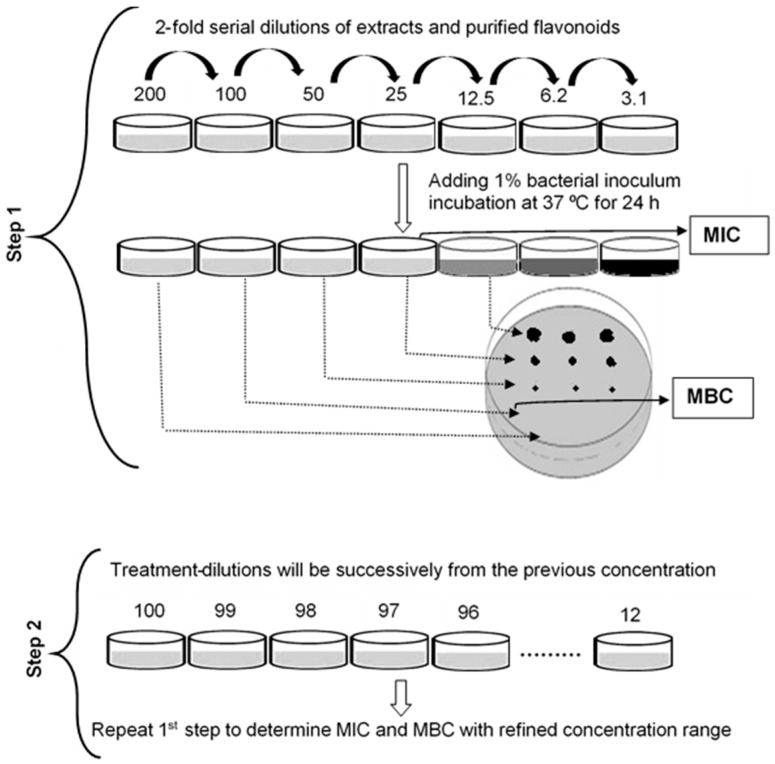
Outline to demonstrate the determination of the MIC and MBC.

**Figure 2 pathogens-12-00657-f002:**
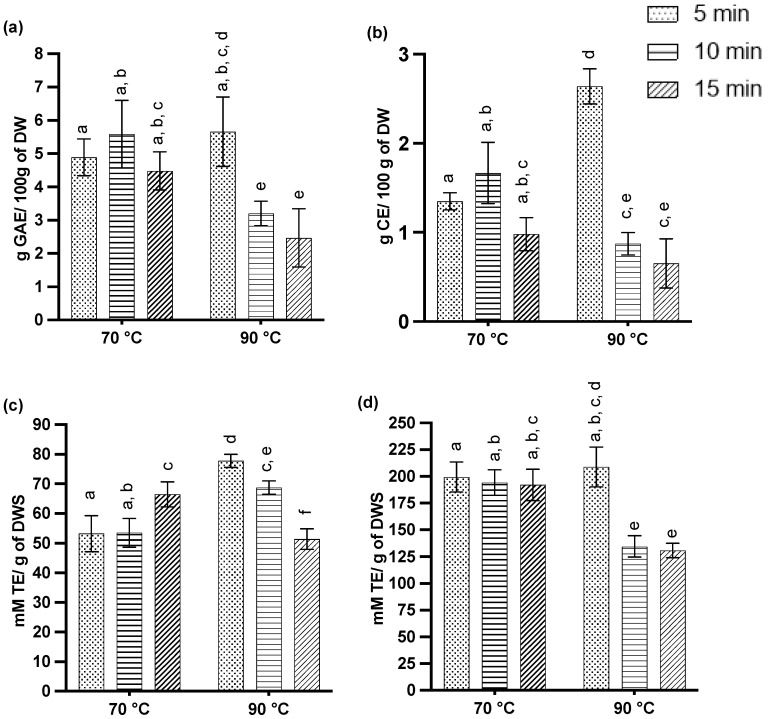
Characterization of sweet orange waste extracts for total polyphenol content; (**a**) total flavonoid content, (**b**) radical scavenging activity in ABTS, (**c**) and ORAC, (**d**) assays. The data represent mean values with standard deviation. Different letters indicate significant differences (*p* < 0.05).

**Figure 3 pathogens-12-00657-f003:**
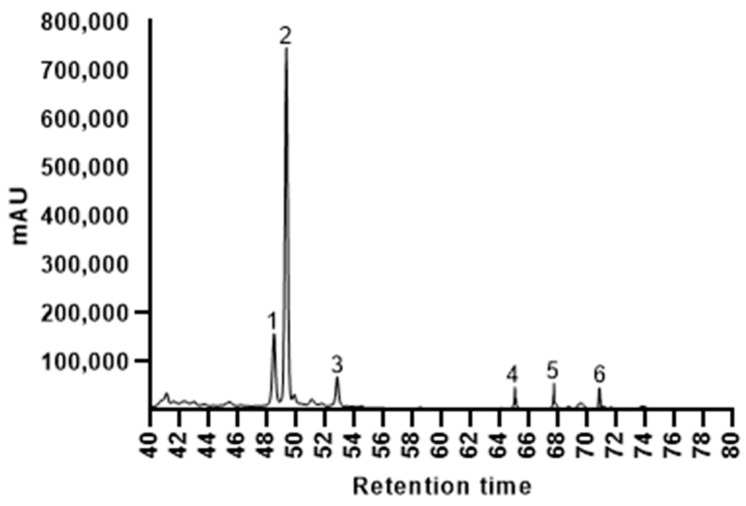
Representative HPLC chromatogram to determine the presence and quantity of flavonoids in the citrus extract samples. Detected flavonoids: 1—narirutin, 2—hesperidin, 3—quercetin, 4—sinensetin, 5—nobiletin, 6—tangeretin.

**Figure 4 pathogens-12-00657-f004:**
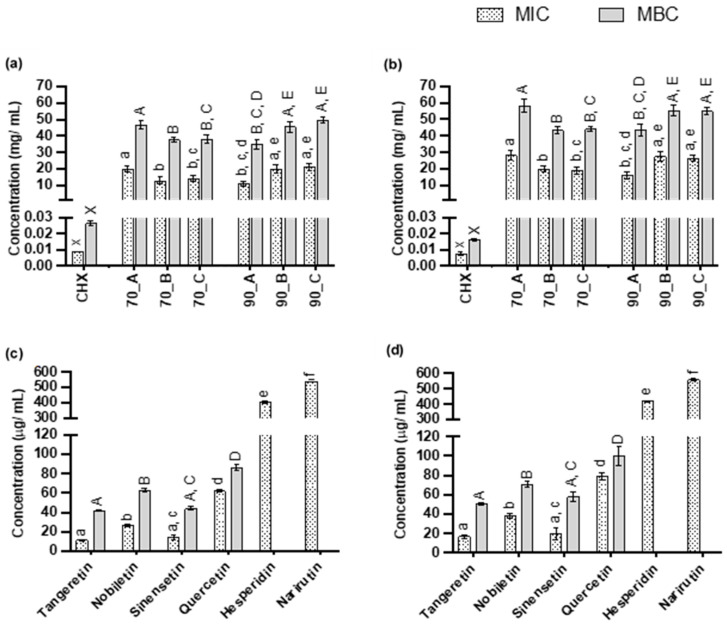
Antibacterial properties of the ISOWE and the individual pure flavonoid compounds against *S. mutans* (**a**,**c**) and *L. casei* (**b**,**d**), determined as the MIC and MBC. The data represent the mean with the standard deviation. The lower case and upper letters case indicate the significant differences (*p* < 0.05) for the MIC and MBC, respectively.

**Figure 5 pathogens-12-00657-f005:**
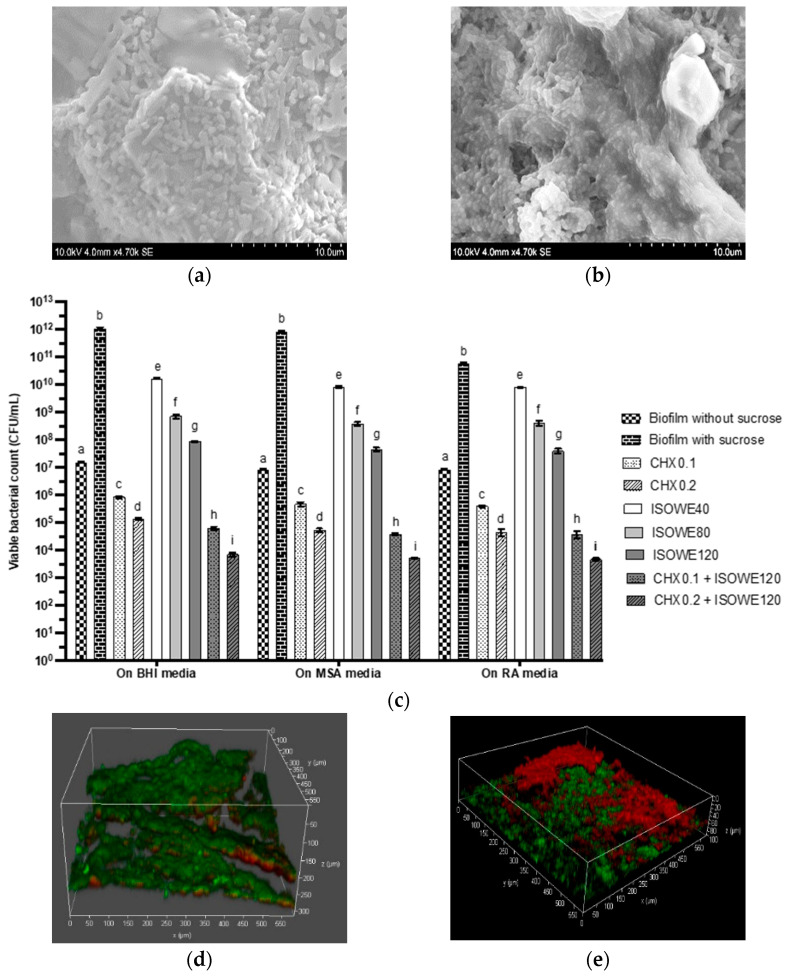
Efficacy of chlorhexidine and ISOWE, alone and in combination, on cariogenic dual-species biofilm formation. The biofilm architecture was assessed using scanning electron microscopy in the absence (**a**) and presence (**b**) of sucrose. Viable bacterial counts (**c**) were obtained for *S. mutans* (MSA medium), *L. casei* (RA medium) and both combined (BHI medium), which were presented as mean with standard deviation. Different letters indicate significant differences within each medium condition (*p* < 0.05). The cariogenic biofilm was imaged using CLSM to indicate live/dead bacterial cells (green/red stain) in the control biofilm (**d**) and under the conditions of treatment with 0.1 (**e**) and 0.2% (**f**) CHX, 40 (**g**), 80 (**h**), 120 (**i**) mg/mL of ISOWE, and the combinations of 120 mg/mL of ISOWE with 0.1% CHX (**j**), and 120 mg/mL of ISOWE with 0.2% CHX (**k**).

**Table 1 pathogens-12-00657-t001:** Quantification of flavonoids in the ISOWE samples (g/100 g dry matter).

Flavonoid	70 °C_A	70 °C_B	70 °C_C	90 °C_A	90 °C_B	90 °C_C
Hesperidin	2.69 ± 0.29 ^a^	2.76 ± 0.30 ^a^	1.42 ± 0.17 ^b^	1.75 ± 0.31 ^b^	1.59 ± 0.31 ^b^	1.74 ± 0.12 ^b^
Narirutin	0.47 ± 0.16 ^a^	0.44 ± 0.13 ^a^	0.34 ± 0.08 ^a^	0.44 ± 0.21 ^a^	0.26 ± 0.08 ^a^	0.26 ± 0.04 ^a^
Quercetin	0.18 ± 0.04 ^a^	0.18 ± 0.04 ^a^	0.17 ± 0.03 ^a^	0.16 ± 0.03 ^a^	0.16 ± 0.04 ^a^	0.19 ± 0.02 ^a^
Sinensetin	0.12 ± 0.03 ^a^	0.11 ± 0.03 ^a^	0.03 ± 0.02 ^b^	0.10 ± 0.02 ^a^	0.01 ± 0.01 ^b^	0.03 ± 0.02 ^b^
Nobiletin	0.09 ± 0.03 ^a^	0.11 ± 0.02 ^a^	0.09 ± 0.02 ^a^	0.10 ± 0.04 ^a^	ND	ND
Tangeretin	0.10 ± 0.04 ^a^	0.08 ± 0.01 ^a^	0.11 ± 0.03 ^a^	0.10 ± 0.02 ^a^	ND	ND

The data represent the mean values with the standard deviation. The different letters indicate the statistical differences (*p* < 0.05) for each flavonoid at different extraction parameters. ND, not detectable.

## Data Availability

The data is contained within the article.

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
