# Peer review of "Antibiofilm Efficacies of Flavonoid-Rich Sweet Orange Waste Extract against Dual-Species Biofilms"

_pathogens, 2023, doi:10.3390/pathogens12050657_

Round 1

Reviewer 1 Report

The study evaluated the antibacterial properties of industrial sweet orange waste  extracts (ISOWE), which are a rich source of flavonoids. Antibacterial activity towards dental cariogenic pathogens Streptococcus mutans and Lactobacillus casei was found. This activity was observed towards planktonic bacteria, as well as for biofilm production and preformed 7-day dual-species biofilms. The ISOW extract also demonstrated strong synergistic effect with chlorgexidine and certain purified flavones (nobletin, tangeretin and sinensetin) were the most effective. The study demonstrates the potential of citrus waste, as a source of flavonoids, for antimicrobial applications in dental health.

The Article is well written, the goal of the study is clear, Introduction uses many references and describes the modern knowledge in this field.

The methods are described carefully, the experiments don’t induce any doubt, Conclusions are in accordance with the results obtained. The Figures and  Tables are understandable and sufficient. The work has not only scientific significance, but may be useful proposal for developing products for dental health.

The Article may be accepted for publishing.

The only small misprints are:

Line 408 – Yersinia enterocolitica

        553 --  sinensetin 

Author Response

The only small misprints are: 

L408 – Yersinia enterocolitica 

Done

L553 --  sinensetin  

Done

Reviewer 2 Report

The article entitled Antibiofilm Efficacies of Flavonoid-Rich Sweet Orange Waste 2 Extract Against Dual Species Biofilms contains important analysis of natural compounds extracted form Citrus spp..

the study is well-designed and the results and discussion are of a high scienific value.

My only remark concerns materials and methods:

L158: Why did authors use  DMSO in tested solitions?

Did authors determine if such a concentrations have its own antibacterial activity? Please add this information into the manuscript.

Author Response

The respective DMSO concentrations were required to dissolve the test compounds. Both positive and negative controls were prepared with the highest concentration of DMSO to confirm the antimicrobial effects were solely due to the test compounds. This information was added to the manuscript, see L160-163.

Reviewer 3 Report

Overall, it is well written article which demonstrate the potential use of citrus waste extracts against cariogenic bacteria and their ability to form biofilms. Methods are well described and results are significant. I will support the publication of the manuscript after minor revisions 

1.Line 21  being on average more than 10 times more effective  need better scientific writing. 

2. Which conc. range was used for MICs calculation

3. What about antimicrobial activity of DMSO. Was it taken in account?

Author Response

1.Line 21  being on average more than 10 times more effective need better scientific writing. :

We have amended the text to: “Citrus flavonoids contributed differently to these effects, with flavones (nobiletin, tangeretin and sinensetin) showing significantly lower MIC and MBC, in comparison to the flavanones hesperidin and narirutin”.

  1. Which conc. range was used for MICs calculation:

The concentrations used to determine MIC and MBC were 1000 – 1 µg/mL, 250 – 5 mg/ mL, and 2000 - 1 µg/mL for CHX, ISOWE, and pure flavonoid compounds respectively.

  1. What about antimicrobial activity of DMSO. Was it taken in account?

The following was added to L160-163: The respective DMSO concentrations were required to dissolve the tested compounds. In addition, both positive and negative controls were prepared with the highest concentration of DMSO to confirm the antimicrobial effects were solely due to the tested compounds.